# Genetic Diversity of Merozoite Surface Antigens in Global *Babesia bovis* Populations

**DOI:** 10.3390/genes14101936

**Published:** 2023-10-13

**Authors:** El-Sayed El-Alfy, Ibrahim Abbas, Rana Elseadawy, Shimaa Abd El-Salam El-Sayed, Mohamed Abdo Rizk

**Affiliations:** 1Parasitology Department, Faculty of Veterinary Medicine, Mansoura University, Mansoura 35516, Egypt; sydnabil@mans.edu.eg (E.-S.E.-A.); ielsayed@mans.edu.eg (I.A.); ranamagdy6666@gmail.com (R.E.); 2Department of Biochemistry and Chemistry of Nutrition, Faculty of Veterinary Medicine, Mansoura University, Mansoura 35516, Egypt; shimaa_@mans.edu.eg; 3Department of Internal Medicine, Infectious and Fish Diseases, Faculty of Veterinary Medicine, Mansoura University, Mansoura 35516, Egypt

**Keywords:** *Babesia bovis*, MSA-1, MSA-2a1, MSA-2b, MSA-2c, genetic diversity, population structure

## Abstract

Cattle can be severely infected with the tick-borne protozoa *Babesia bovis*, giving rise to serious economic losses. Invasion of the host’s RBCs by the parasite merozoite/sporozoites depends largely on the MSA (merozoite surface antigens) gene family, which comprises various fragments, e.g., MSA-1, MSA-2a1, MSA-2a2, MSA-2b and MSA-2c, highlighting the importance of these antigens as vaccine candidates. However, experimental trials documented the failure of some developed MSA-based vaccines to fully protect animals from *B. bovis* infection. One reason for this failure may be related to the genetic structure of the parasite. In the present study, all MSA-sequenced *B. bovis* isolates on the GenBank were collected and subjected to various analyses to evaluate their genetic diversity and population structure. The analyses were conducted on 199 MSA-1, 24 MSA-2a1, 193 MSA-2b and 148 MSA-2c isolates from geographically diverse regions. All these fragments displayed high nucleotide and haplotype diversities, but the MSA-1 was the most hypervariable and had the lowest inter- and intra-population gene flow values. This fragment also displayed a strong positive selection when testing its isolates for the natural selection, which suggests the potential occurrence of more genetic variations. On the contrary, the MSA-2c was the most conserved in comparison to the other fragments, and displayed the highest inter- and intra-population gene flow values, which was evidenced by a significantly negative selection and negative neutrality indices (Fu’s Fs and Tajima’s D). The majority of the MSA-2c tested isolates had two conserved amino acid repeats, and earlier reports have found these repeats to be highly immunogenic, which underlines the importance of this fragment in developing vaccines against *B. bovis*. Results of the MSA-2a1 analyses were also promising, but many more MSA-2a1 sequenced isolates are required to validating this assumption. The genetic analyses conducted for the MSA-2b fragment displayed borderline values when compared to the other fragments.

## 1. Introduction

Tick-borne diseases impede the growth of the worldwide cattle industry and cost farmers billions of dollars each year [1,2]. Various tick-borne *Babesia* and *Theileria* species can cause life-threatening infections giving rise to serious economic consequences [3,4,5]. Bovine babesiosis is economically significant in terms of the potential production losses (e.g., deaths, abortions, decreased meat and milk output) and expensive control measures other than the restrictions enforced due to the effects this disease on the cattle trade [1,2]. *B. bovis*, *Babesia bigemina* and *Babesia divergens* are the main species that cause bovine babesiosis, with *B. bovis* infections being associated with more severe disease and higher mortality, which makes this species the most virulent [3,6,7,8].

Variable merozoite surface antigens (VMSA) of *B. bovis* are surface-exposed glycoproteins that are attached to the cell membrane via glycosyl-phosphatidylinositol (GPI) moieties [9,10,11]. The variable merozoite surface antigen (VMSA) gene’s family is made up of the merozoite surface antigen 1 (MSA-1) and MSA-2, which encode 42- and 44-kDa proteins, respectively, as well as babr genes [10]. The MSA-1 is a single copy gene [12] whereas, the locus 2 contains four tandemly organized copies of MSA-2-related genes; MSA-2a1, MSA-2a2, MSA-2b and MSA-2c [11]. An earlier study suggested that the immunodominant 42- and 44-kDa merozoite polypeptide epitopes are not conserved across *B. bovis* strains from different geographic areas [13]. In addition, the antigenic polymorphism of the MSA-1 is demonstrated between *B. bovis* strains as a result of different alleles encoding distinctly different proteins [12].

Based on their surface location and neutralizing activity, the VMSA members are thought to play key functions in the invasion of RBCs by the parasite merozoite and tick-derived sporozoites [10,12,14,15,16]. The variability of the MSA genes allows the *B. bovis* parasites to elude the host’s defense mechanism [17]. Various Epidemiological surveys have been conducted to evaluate the genetic situation of *B. bovis*. These surveys have mostly recruited a few numbers of *B. bovis* field isolates from different countries, and were based on the phylogenetic analysis employing a single or at most double MSA fragment, particularly the MSA-1, and occasionally the MSA-2b and MSA-2c. Nevertheless, only one study conducted haplotype analysis on MSA-2b and MSA-2c gene sequences based on a few numbers of isolates (33 and 37, respectively) [18]. Therefore, the present study aimed for a comprehensive genetic analysis of all published nucleotide sequences of the global *B. bovis* MSA-tested isolates. This analysis would be beneficial in expanding the current knowledge in order to use these gene fragments as molecular markers in future epidemiological studies. Another objective of this study is to determine the conserved peptides in the MSA gene that might be useful as antigen and vaccine candidates.

## 2. Methods

### 2.1. Collection of B. bovis-MSA Nucleotide Sequences

The National Centre for Biotechnology’s website (http://www.ncbi.nlm.nih.gov accessed on 13 June 2023) was searched in June 2023 to retrieve the nucleotide sequences and the corresponding amino acid sequences of *B. bovis* merozoite surface antigens that had been deposited in the GenBank from various countries. The term *B. bovis* was used in various combinations with the following key words: MSA-1, MSA-2a1, MSA-2a2, MSA-2b and MSA-2c. All *B. bovis*-MSA nucleotide sequences were collected, but those shorter than 500 bp were excluded from our analysis. In addition, published articles citing the accession numbers were retrieved. Data describing the collected sequences were extracted and organized in a Microsoft Excel^®^ spreadsheet. The following information was extracted: GenBank accession numbers, isolate country and hosts, authors and publication year (Appendix A). The collected nucleotide and amino acid sequences were aligned using the Clustal Omega Multiple Sequence Alignment tool on the EMBL-EBI server [19].

### 2.2. Phylogenetic and Population Structure Analyses

The pairwise distances among isolates of each MSA fragment were calculated using the software MEGA X, and the same software was used to calculate the *d_N_*/*d_S_* (relationship between the amount of non-synonymous substitutions per non-synonymous site and the amount of synonymous substitutions per synonymous site), employing the codon-based Z test for selection based on the modified Nei–Gojobori method (Jukes–Cantor correction). The MEGA X was also used to construct phylogenetic trees for the aligned nucleotide sequences employing maximum likelihood phylogeny analysis based on Hasegawa–Kishino–Yano (HKY) Model. The phylogenetic trees were then optimized using the online tool ITOL V6 [20]. MEGA (7) was then used to convert the nucleotide sequences into Nexus format [21], which was used to establish the haplotype networks using the Minimum Spanning Model of PopArt 1.7. Various haplotype networks were established for isolates of each MSA fragment in relation to type of the host (cattle, water buffalo and tick vectors), as well as the geographic locations including the country and the continent [22].

The population structure analyses were conducted using the software DnaSP6 [23]. The collected nucleotide sequences for each MSA fragment were grouped according to their hosts, country and continent. Values for various diversity, neutrality and fixation indices were calculated including: numbers of polymorphic sites (S), total number of mutations (Eta), nucleotide diversity (π), average number of pairwise nucleotide differences (k), haplotypes number (H), haplotype diversity (Hd), sequence conservation (C), Tajima’s D [24], Fu’s Fs [25], pairwise genetic difference (Fst), genetic differentiation index based on the frequency of haplotypes (Gst), average number of pairwise nucleotide differences (Kxy), gene flow (Nm), nucleotide substitution per site (Dxy) and net nucleotide substitution per site (Da) between various groups (populations).

## 3. Results

A total of 594 partial nucleotide sequences were retrieved during the GenBank search for *B. bovis* isolates that had been tested for various MSA gene (MSA-1, MSA-2a1, MSA-2b and MSA-2c) fragments (Appendix A). The majority of these sequences were published in studies (n = 22). A few isolates (n = 8) for the MSA-2a2 fragment were found deposited in the GenBank, thus this fragment was defined unsuitable for conducting the analysis. The alignment of the MSA-1 nucleotide sequences revealed multiple insertion–deletion polymorphisms (INDELs) throughout the reading frames as well as single nucleotide polymorphisms (SNPs). Therefore, the genetic diversity indices were based on both INDELs and SNPs without further trimming of the aligned sequences. On the contrary, the alignment of the MSA-2a1, MSA-2b and MSA-2c partial gene sequences revealed INDELs at the both ends only. Thus, their sequences were trimmed from both sides (723 bp, 408 bp, 618 bp, respectively) and the diversity indices were based only on single nucleotide polymorphisms (SNPs).

### 3.1. MSA-1 Isolates

A total of 199 *B. bovis* isolates tested for the MSA-1 fragment were revealed during our database search and were used for the analyses conducted in the present study. All of these isolates belonged to cattle from various continents, but not Europe. No isolates from water buffalo or the tick vectors were found. After alignment, these isolates yielded a sequence length of 1014 bp with 505 polymorphic sites and 1014 total mutations, which gave rise to a very high nucleotide diversity value (0.40) in comparison to the other tested fragments. This was also demonstrated by the very low sequence conservation value (0.078) when compared to those of the other tested fragments. In addition, MSA-1 isolates comprised 107 haplotypes with a haplotype diversity of 0.986 (Table 1). When these isolates were tested for natural selection, statistically significant positive selection was revealed, indicating the dominance of nonsynonymous over synonymous substitutions (Table 2). Interestingly, a positive high value for the Fu’s Fs (18.416) index was detected suggesting the recent population bottleneck. Likewise, the Tajima’s D neutrality index displayed a positive, but statistically insignificant, value (Table 1).

Phylogenetically, the 199 MSA-1 isolates comprised three ancestral groups, of which a small group included only 10 isolates mostly from Asia, except a single isolate from Australia. Likewise, the 112 isolates that formed the third ancestral group came mostly from Asia, except two isolates from Brazil. On the contrary, the second ancestral group included 77 isolates from cattle in various countries worldwide, including the remaining Asian, Brazilian and Australian isolates, the African isolates and the Mexican isolates (Appendix A). Notably, the tested isolates included two *B. bovis* vaccinal strains (K and T) that were isolated from cattle in Australia. Both strains were clustered in the second ancestral group, and formed two separate haplotypes, each comprising a single isolate from cattle in Australia beside the vaccinal strain (Appendix A). These phylogenetic patterns suggest population separation which was confirmed via the population structure analysis. When the MSA-1 isolates from various continents were compared, low gene flow (Nm) and high genetic differentiation (Gst) values were revealed, even between isolates from the Americas (North and South) (Appendix A). Country–country comparisons also supported this suggestion; low gene flow values were observed among isolates from various countries even in the same continent. Notably, the African isolates (from Ghana) exhibited the lowest gene flow values compared to the other worldwide isolates (Appendix A).

### 3.2. MSA-2a1 Isolates

Only 24 MSA-2a1 isolates, from four countries (Sri Lanka, Mexico, Argentina, Israel), were revealed during our GenBank search. The length of the aligned sequences was 723 bp with 283 polymorphic sites and 320 total mutations. Isolates of this fragment had the highest sequence conservation value (C = 0.609) in comparison to the three other fragments (Table 1). The MSA-2a1 nucleotide sequences displayed statistically significant, negative selection. However, 23 haplotypes out of the 24 isolates were detected, including a single shared haplotype (Hap-21) comprising two isolates from cattle in Sri Lanka. The haplotype network suggested that the Argentinian isolates and a single isolate from Israel were evolved from this haplotype. The remaining Israeli isolates were clustered with those from Mexico (Figure 1). Nonetheless, comparing isolates from both countries (Israel and Mexico) displayed a low gene flow value (Nm = 5.04). On the contrary, the Israeli isolates showed high gene flow values (Nm = ~34) when compared to those from Argentina or Sri Lanka (Appendix A). Moreover, all tested isolates displayed insignificant negative values for the Fu’s Fs and Tajima’s D neutrality indices (Table 1).

### 3.3. MSA-2b Isolates

A total of 193 isolates tested for the MSA-2b fragment were found during the GenBank search. These isolates belonged mostly to cattle (n = 176), but a few numbers belonged to water buffalo (n = 10) and tick vectors (n = 7). The isolates came from various countries across four continents (Africa, Asia and the Americas). Aligned nucleotide sequences (409 bp in length) of these isolates revealed 283 polymorphic sites as well as 405 total mutations, giving rise to a nucleotide diversity of 0.18, which was around half that of the MSA-1 fragment. On the contrary, the sequence conservation value for the MSA-2b tested isolates (0.287) was much higher than that of the MSA-1 tested isolates (0.065); nonetheless, both fragments displayed positive selection (Table 2). The 193 MSA-2b isolates varied genetically by 0.0 to 59.0% (pairwise distance), and comprised 104 haplotypes giving rise to 0.981 haplotype diversity value (Table 1). Of the 104 haplotypes, 27 were shared and included 2–15 isolates.

No specific patterns were revealed for distribution of certain haplotypes in relation to the host or the country, i.e., isolates from various countries were intermingled with each other in various clades (Appendix A). For the hosts, the MSA-2b haplotypes that belong to water buffaloes or tick vectors were evolved from those of cattle. In addition, some water buffalo isolates shared two haplotypes (H15 and H41) with those from cattle, suggesting their phylogenetic association (Figure 2).

The phylogenetic analysis also illustrated circulation of a few haplotypes among the sampled continents, but particularly Asia and Africa (Figure 3). For example, seven out of ten haplotypes that circulated among various continents included isolates from Africa and Asia, and four out of these ten haplotypes circulated between Asia and North America (Figure 3).

This would account for the high gene flow values between isolates from Asia when compared to those from Africa (Nm = 28.51) and North America (Nm = 32.93) (Appendix A). Interestingly, the lowest gene flow value (Nm = 13.78) was detected when comparing isolates from the Americas (North and South), despite their very close geographic borders. This was also the case when comparing isolates from two neighboring African countries: South Africa and Mozambique; a very low gene flow value (Nm = 6.77) was detected. However, the former isolates expressed a high gene flow value (35.42) compared to those from a country outside Africa (China). Likewise, isolates from Sri Lanka, located in Asia, displayed high gene flow value (43.21) when compared to those from Mexico, located in North America. On the other hand, the China–Philippines comparison provides the only example for the high gene flow between isolates from two countries in the same continent (Appendix A).

### 3.4. MSA-2c Isolates

During the GenBank database search, a total of 148 *B. bovis* isolates tested for the MSA-2c fragment were revealed. The majority (n = 134) of these isolates belonged to cattle from various countries across four continents (Asia, Europe, and the Americas). A few isolates from water buffalo (n = 11) and tick vectors (n = 3) were also found. Nucleotide sequences of these isolates displayed 409 polymorphic sites and 546 total mutations in 618 bp aligned nucleotide sections, giving rise to the lowest nucleotide diversity value (0.09648) among all tested MSA fragments (Table 1). In addition, a strong negative selection was hypothesized after testing the MSA-2c isolates for natural selection, suggesting the dominance of synonymous over nonsynonymous substitutions (Table 2). Despite having a higher sequence conservation value (0.333) than the MSA-2b (0.287) or MSA-1 (0.078) fragments, this fragment also had a high haplotype diversity value (0.995); the 148 MSA-2c isolates formed 122 haplotypes, which varied genetically by 0.0–50.2% (Table 1). 

Similarly to the MSA-2b isolates, the MSA-2c isolates from water buffalo as well as ticks were evolved from those from cattle, which points to absence of specific patterns for host or country-related distribution of the haplotypes. However, among the 148 isolates, 10 (represented by haplotypes # 10, 11, H31–H34, and H36–H37) from cattle in Brazil, the Philippines, China and Sri Lanka, as well as an isolate from ticks in Israel (H35) showed considerable identity between each other, but greatly varied genetically from the remaining isolates (Figure 4).

At the continent level, there was no specific pattern for the distribution of various MSA-2c haplotypes in various continents (Figure 5). In addition, the inter-continental comparisons revealed higher gene flow and lower genetic differentiation values for the MSA-2c isolates against those of the MSA-1 or MSA-2b isolates (Appendix A). In contrast to the MSA-2b isolates, comparing the MSA-2c isolates from the Americas revealed a very high gene flow value (176.75). This was clearly demonstrated when comparing isolates from Mexico (North America) and Brazil (South America); a very high gene flow value (1858.72) was estimated (Appendix A). In addition, high gene flow values were also detected when comparing isolates from various countries in Asia, e.g., Vietnam–Philippines (Nm = 52.87), Mongolia–Israel (Nm = 90.00), and Philippines–Sri Lanka (Nm = 59.84). Notably, the European isolates from Turkey displayed slightly high gene flow values when they were compared to isolates not only from geographically related countries (China and Mongolia), but also to other countries worldwide (e.g., Mexico and Brazil). The inter- and intra-continental high gene flow values among the MSA-2c isolates (Appendix A) in comparison to those of the other MSA fragments were evidenced by a significantly negative estimated value for the Fu’s Fs index (−39.751), suggesting population expansion (Table 1).

### 3.5. Detecting the Conserved Regions in the Aligned Amino Acid Sequences of Various MSA Gene Fragments

Three short repeats, each consisting of 20 amino acids (aa) in the MSA-1 fragment, (PEGSFYDDMSKFYGAVGSFD, NALIKNNPMIRPDLFNATIV, TDIVEEDREKAVEYFKKHVY) were detected within 41 isolates, specifically in 36, 31 and 38 out of the 199 MSA-1 isolates, respectively. It is noteworthy that these isolates came from various countries including Ghana, Mongolia, Thailand, Sri Lanka, Mexico and Brazil. For the MSA-2c fragment, three aa repeats with variable lengths were noticed. Out of the 148 MSA-2c tested isolates, 114 contained the 23 aa motif “HDALKAVKQLIKTDAPFNTSDFDT”, and 131 contained the 7 aa motif “FINPSST”. On the other hand, four conserved regions in the MSA-2a1 as well as the MSA-2b fragments were evaluated. Out of the 193 MSA-2b isolates tested, 96 had the ten aa motif “EFNAFLNDNP”, 117 had the four aa motif “YYKK” and 169 had the six aa motif “VKFCND”. These motifs were also found in 11, 12 and 23 of the 24 MSA-2a1 isolates, respectively. However, all MSA-2a1 and MSA-2b isolates had a four aa motif “SPFM”.

## 4. Discussion

*B. bovis* has long been known to have different strains and subpopulations based on a variety of molecular markers [26,27]. These markers include the merozoite surface antigen (MSA), a protein-coding gene that has been widely used and displays high genetic diversity among *B. bovis* isolates in various geographical regions and hosts, predominately including cattle, but also water buffalo and tick vectors. A few other genetic markers were occasionally used, but displayed low genetic diversity levels, e.g., the 18S rRNA, rhoptry-associated protein 1 (RAP-1a), Spherical Body Protein-2 (sbp-2), Spherical Body Protein-4 (sbp-4) and the Thrombospondin-Related Anonymous Protein (trap) [5,27,28,29]. Since the MSA gene is very important for RBC invasion by *Babesia* merozoites/sporozoites, and there have been many *B. bovis* isolates tested for this gene, the present study provides the first comprehensive genetic analysis for all partial nucleotide sequences of *B. bovis*-MSA-tested isolates published on the GenBank from 22 studies [11,18,30,31,32,33,34,35,36,37,38,39,40,41,42,43,44,45,46,47,48,49]. These isolates represent various MSA gene fragments including the MSA-1 (n = 199 isolates), MSA-2a1 (n = 24), MSA-2b (n = 193) and MSA-2c (n = 148). Several intra- and inter-fragmental genetic comparisons were conducted, and all have confirmed the very high genetic diversity of this gene family.

The MSA-1 nucleotide sequences displayed the greatest nucleotide diversity and the lowest sequence conservation in comparison to sequences from the other MSA gene fragments, most likely due to the multiple polymorphisms, either INDELs or SNPs, throughout the MSA-1 reading frames. Earlier reports have also documented greater sequence diversity among the MSA-1 field isolates, as well as extensive sequence variation in all MSA-1 vaccination breakthrough isolates [38,50,51]. This is particularly important because a great polymorphism may result in a partial-to-complete absence of MSA-1 B- and T-lymphocyte cross-reactivity among *B. bovis* strains [12,13,52,53,54]. This was evidenced by the failure of the recombinant MSA-1 vaccination to provide immunity against a virulent *B. bovis* (T2Bo strain) challenge [54], raising concerns about the validity of the MSA-1 gene fragment as a vaccine candidate. In addition, the MSA-1 isolates analyzed in the present study displayed strong positive selection when tested for the natural selection, indicating the dominance of nonsynonymous over synonymous substitutions. Non-synonymous mutations can modify the amino acid sequence of the protein encoded, whereas synonymous mutations are functionally silent and assumed not to alter the encoded protein [55,56]. Positive selection can therefore cause genetic variations to become more prevalent or to remain in the population. On the contrary, negative selection can lessen the existence of genetic variants or eliminate them from the population [57]. Similarly, strong positive selection has also been documented in the highly immunogenic RhopH protein of Plasmodium falciparum [58]. This shows that these parasites are selected based on their capacity to evade host recognition [59], and hypothesizes the limited usefulness for such positively selected genes in vaccination against these parasites. However, Cuy-Chaparro et al. [60] have identified three MSA-1 non-overlapping peptides derived from functionally constrained regions that can specifically bind to a sialoglycoprotein located on the surface of bovine erythrocytes, two of these peptides exhibit a helical structure and conserved patterns across all MSA-1 strains/isolates tested (n = 37). When these peptides (PEGSFYDDMSKFYGAVGSFD, NALIKNNPMIRPDLFNATIV, TDIVEEDREKAVEYFKKHVY) were searched within the aligned protein sequences of the 199 MSA-1 isolates analyzed in the present study, they were detected within only 41 isolates, which suggests a lack of conserved regions in the hypervariable MSA-1 gene locus across the global *B. bovis* population. Moreover, the 41 isolates came from various countries (e.g., Ghana, Mongolia, Thailand, Sri Lanka, Mexico and Brazil), and these conserved amino acid motifs were never detected in all the isolates from a certain country, suggesting also a lack of country-based selection in vaccine targets.

On the other hand, the MSA locus 2 contains four tandemly arranged copies of MSA-2-related genes including the MSA-2a1, MSA-2a2, MSA-2b and MSA-2c [11]. Three (MSA-2a1, MSA-2b, and MSA-2c) of them were analyzed in the present study and displayed lower genetic diversities in comparison to that of the MSA-1 gene. Relatively lower genetic variations have also been documented among the MSA-2 genes, either in the vaccinal strains or breakthrough isolates in comparison to those of the MSA-1 [61]. The MSA-2a1 and -2a2 have the most similar protein structure with around 90% identity and only two different regions [11]. The differences are represented by a 24-amino acid repeat, which is present once in MSA-2a2 but duplicated in MSA-2a1. In addition, there is a 32-amino acid recombination region that differs between MSA-2a1 and -2a2, but is identical when comparing MSA-2a2 to MSA-2b. This region is surrounded by strictly conserved segments of 10 (EFNAFLNDNP) and 6 (VKFCND) amino acids, and is thought to be highly immunogenic [11]. We have observed both motifs in the aligned amino acid sequences of many MSA-2a1 and MSA-2b isolates analyzed in the present study. The second one (VKFCND), in particular, was detected in 23 and 169 out of the 24 MSA-2a1 and 196 MSA-2b tested isolates. This recombination area also includes a conserved 4 aa (YYKK) central motif [11,17] that was also detected in 12 and 117 out of the analyzed isolates for both gene fragments, respectively. Moreover, a different 4 aa motif that has been previously found conserved [61] and occurs in a hydrophilic region, was interestingly identified in all MSA-2a1 and MSA-2b isolates analyzed in the present study. Although many more MSA-2a1 sequenced field isolates are required to confirm their genetic structure, evaluating the antigenic properties of these conserved motifs, including the induction of neutralizing antibodies and cellular Th1 immune response, may be promising and can highlight the role of both gene fragments (MSA-2a1 and MSA-2 b) in the vaccination strategy against *B. bovis*.

Out of the four analyzed gene fragments in the present study, the MSA-2c appears to be the most conserved. The MSA-2c isolates had (1) the lowest nucleotide diversity, (2) the greatest inter- and intra-population gene flow values and (3) high sequence conservation, in comparison to the other gene fragments. However, 11 of the 148 MSA-2c analyzed isolates from various countries were greatly differed from the others, most likely due to the existence of recombination and functional mutations in this gene. The remaining 137 isolates were genetically similar and displayed a lower mean overall genetic distance (*p*-distance = 0.05) in comparison to that of the whole 148 isolates (*p*-distance = 0.10). The inter- and intra-continental high gene flow values among the MSA-2c isolates were evidenced by a significantly negative estimated value for the Fu’s Fs index and insignificant negative Tajima’s D. Low nucleotide diversity and high haplotype diversity together point to a rapid expansion from limited population size which is demonstrated by the clustering of isolates from different countries into major blocks and suggests the usefulness of this marker in genotyping studies. The MSA-2c fragment also exhibited a strong statistically significant negative selection. Once again, negative selection lessens the occurrence of genetic variants, and limited genetic variations are required for successful vaccinations. Thus, the MSA-2c fragment is hypothesized to have a substantial role in developing diagnostics and vaccines against *B. bovis*. This assumption has been confirmed in several studies, which have demonstrated that antibodies against the MSA-2c can neutralize the invasion of erythrocytes via *B. bovis* merozoites [11,14,15]. In addition, the MSA-2c possesses highly immunogenic and conserved B- and T-cell epitopes [15,62]. This makes Bono et al. [63] suggest the MSA-2c recombinant protein as an antigen candidate for the diagnosis of bovine babesiosis caused by *B. bovis*.

Moreover, the presence of functionally restricted regions can result in such negative selection of the MSA-2c fragment. Apart from the 11 genetically different MSA-2c isolates, the remaining 137 isolates displayed two conserved amino acid motifs including the 23 aa motif “HDALKAVKQLIKTDAPFNTSDFDT” that was detected in 114 isolates and the 7 aa motif “FINPSST” that was detected in 131 isolates. Both motifs have interestingly been defined as highly immunogenic [62]. Additionally, the “FINPSST” motif has promising antigenic properties such as generation of neutralizing antibodies in vitro and induction of a higher percentage of activated CD4+/CD45RO+ T cells. This motif also triggers the production of γ interferon (IFN-γ) in PBMCs from vaccinated cattle, indicating the presence of a long-lasting Th1 immune response [62]. However, immunization with the recombinant MSA-1 and MSA-2c proteins expressed from the Mexico RAD *B. bovis* strain was insufficient to prevent clinical symptoms after challenge exposure [64]. Nonetheless, the MSA-1 (accession number EF640955) and MSA-2c (EF640942) fragments of this strain contain the previously mentioned immunogenic regions, which underline the need for many more studies to evaluate the antigenic characters of various MSA proteins.

## 5. Conclusions

This study is the first comprehensive analysis of diversity across *B. bovis* MSA family genes using global population isolates. The MSA-2c is a promising gene in terms of molecular epidemiological studies for clustering isolates from different geographical regions based on sequences diversity. The negative selection of this gene and the presence of highly antigenic and functionally restricted regions such as the 7 aa motif “FINPSST” suggested it to be a suitable candidate for recombinant vaccination along with other *B. bovis* immunogenic gene peptides.

## Figures and Tables

**Figure 1 genes-14-01936-f001:**
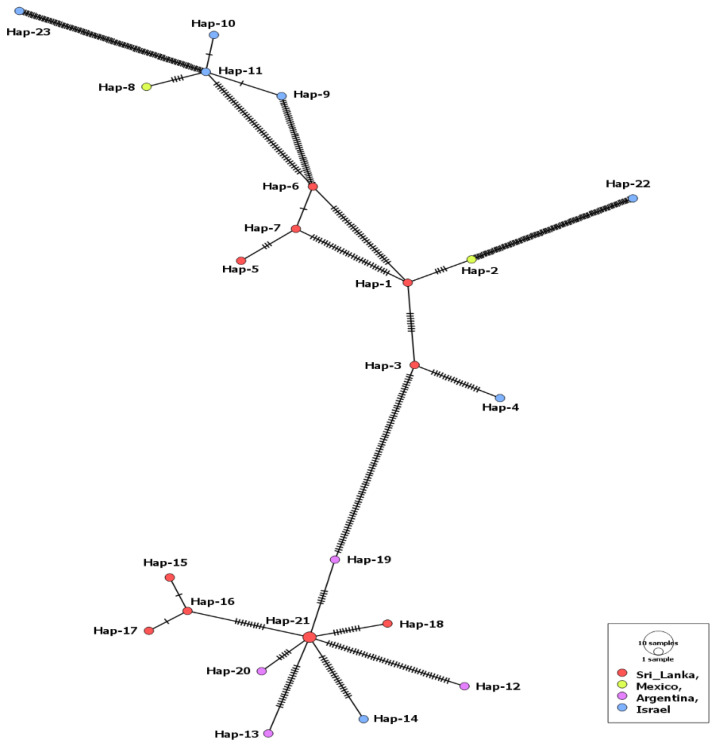
Haplotype network of *B. bovis* MSA-2a1 gene fragment describes the distribution of the revealed haplotypes in relation to countries, which is indicated by different colors. The circle size is consistent with the haplotype frequency. The number of mutations distinguishing the haplotypes is shown by hatch marks.

**Figure 2 genes-14-01936-f002:**
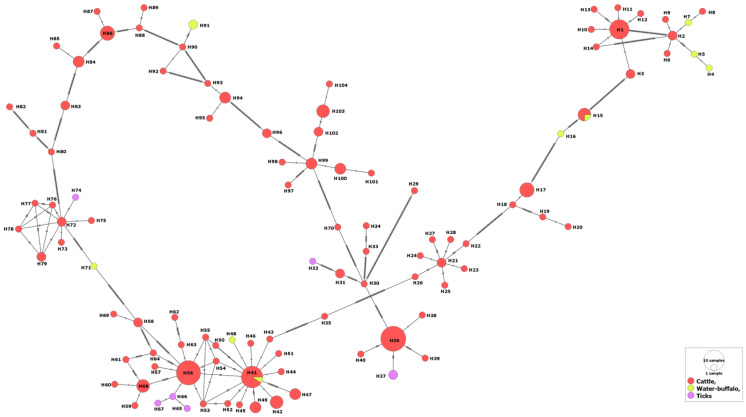
Haplotype network of *B. bovis* MSA-2b gene fragment describes the distribution of the revealed haplotypes in relation to hosts and indicated by different colors.

**Figure 3 genes-14-01936-f003:**
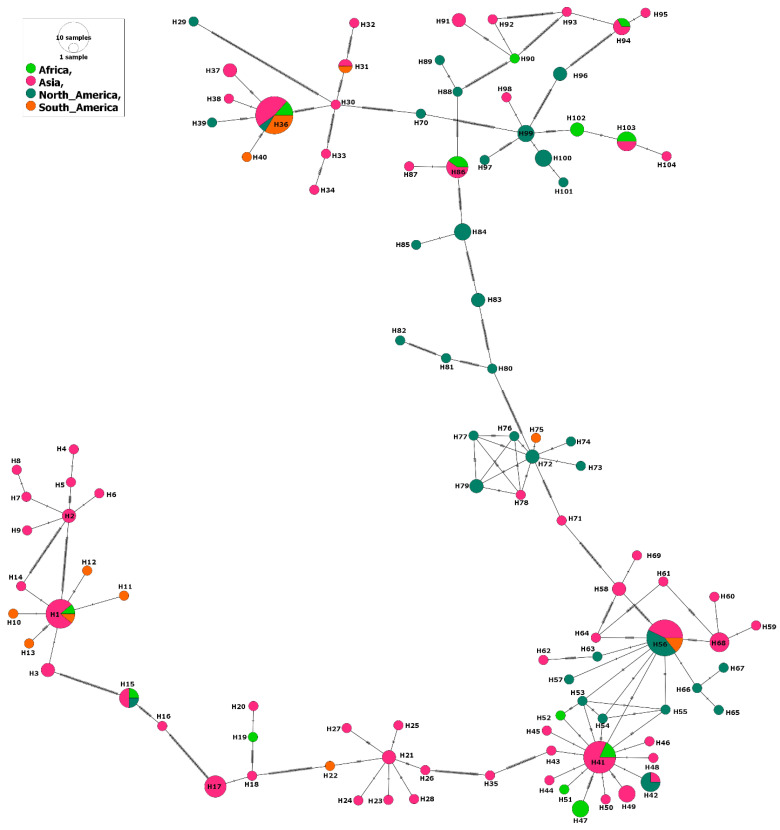
Haplotype network of *B. bovis* MSA-2b gene fragment describes the distribution of the revealed haplotypes in relation to continents and indicated by different colors.

**Figure 4 genes-14-01936-f004:**
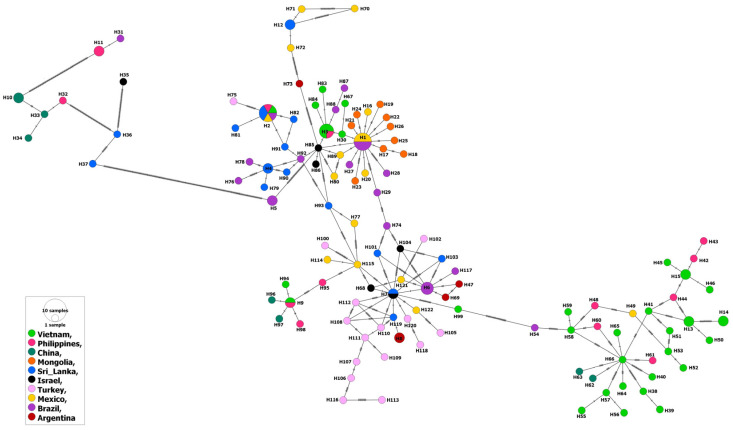
Haplotype network of *B. bovis* MSA-2c gene fragment describes the distribution of the revealed haplotypes in relation to countries and indicated by different colors.

**Figure 5 genes-14-01936-f005:**
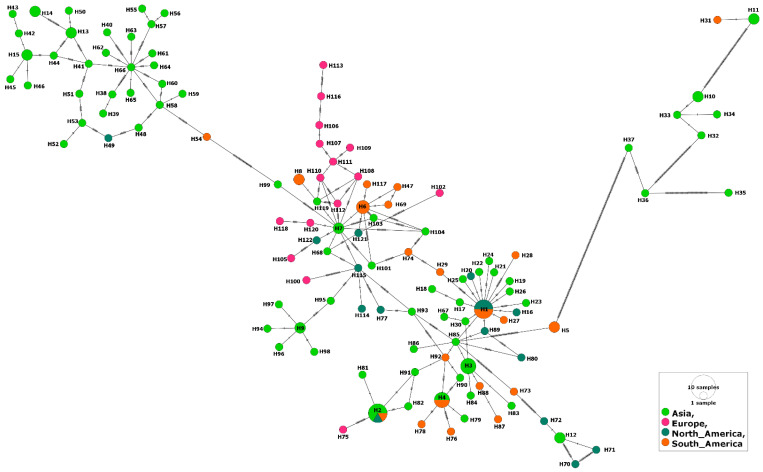
Haplotype network of *B. bovis* MSA-2c gene fragments describes the distribution of the revealed haplotypes in relation to continents and indicated by different colors.

**Table 1 genes-14-01936-t001:** Diversity indices of various *B. bovis* isolates tested at four fragments of the MSA gene.

Gene Fragment	n	L	S	Eta	K	π	H	Hd	C	Fu’s Fs Statistic	Tajima’s D	
											Value	Significance
MSA-1	199	1014	505	1014	219.74255	0.40246	107	0.986	0.078	18.416	0.87536	>0.10
MSA-2a1	24	723	283	320	82.93478	0.11471	23	0.996	0.609	−1.358	−0.12975	>0.10
MSA-2b	193	408	283	409	71.93469	0.18074	104	0.981	0.287	−2.543	0.08555	>0.10
MSA-2c	148	618	409	546	59.23727	0.09648	122	0.995	0.333	−39.751	−1.29882	>0.10

n, number of sequences tested; L, base pair length of the aligned sequences; S, number of segregating (polymorphic/variable) sites; Eta, total number of mutations; π, nucleotide diversity; K, average number of pairwise nucleotide differences; H, number of haplotypes; Hd, haplotype diversity; C, sequence conservation.

**Table 2 genes-14-01936-t002:** Natural selection values for the tested nucleotide sequences of various MSA fragments of *B. bovis*.

Gene Fragment	Neutral Selection (HA: *d_N_* ≠ *d_S_*)		Positive Selection (HA: *d_N_* > *d_S_*)		Negative Selection (HA: *d_N_* < *d_S_*)	
	Value	*p*-Value	Value	*p*-Value	Value	*p*-Value
Msa-1	8.29	0.00	8.01	0.00	−8.19	1.00
Msa-2a1	−2.24	0.03	−2.25	1.00	2.27	0.01
Msa-2b	1.62	0.11	1.59	0.06	−1.59	1.00
Msa-2c	−3.21	0.00	−3.21	1.00	3.15	0.00

HA, Hypothesis alternative; *d_N_*: nonsynonymous substitutions. *d_S_*: synonymous substitutions. Values are statistically significant when *p*-value < 0.05.

## Data Availability

All data generated or analyzed during this study are included in this published article and its Appendix A.

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
