# Peer review of "Genetic Diversity of Merozoite Surface Antigens in Global Babesia bovis Populations"

_genes, 2023, doi:10.3390/genes14101936_

Round 1

Reviewer 1 Report

This study describes a comprehensive analysis of the genetic diversity of Babesia bovis merozoite surface antigen (MSA) gene families using GenBank sequences collected from across the world. The research showed that most of these genes have high genetic diversity globally, making them generally poor antigens for vaccine development. Only a single family was found to be conserved and considered a suitable vaccine candidate. This work is significant for the search for vaccine candidates against this economically important disease.

line 253: should say H31-H34

The language would benefit from copy-editing by a native speaker. Although it is understandable and mostly reads well, there are consistent grammatical mistakes throughout, including unnecessary use of "the" and use of the incorrect tense, e.g. line 39 should say "evaluate", and multiple times the word "consisted" is used rather than "consisting" etc.

Author Response

Response to the Comments from the editor and reviewers

Manuscript ID: genes-2593345

Title: Genetic diversity of merozoite surface antigens in the global Babesia bovis populations

Comments from the editors and reviewers: 

First, we thank the reviewers for providing constructive comments and suggestions, which were useful for improving the quality of the manuscript. We have now modified our manuscript in acknowledgement of all of the reviewers’ comments and suggestions. In the following pages are our point-by-point responses to each of the comments. The changes made to the manuscript are marked up using the “Track Changes” function. All references are relevant to contents of the revised version of the manuscript.

Reviewer-1

Comments and Suggestions for Authors:

This study describes a comprehensive analysis of the genetic diversity of Babesia bovis merozoite surface antigen (MSA) gene families using GenBank sequences collected from across the world. The research showed that most of these genes have high genetic diversity globally, making them generally poor antigens for vaccine development. Only a single family was found to be conserved and considered a suitable vaccine candidate. This work is significant for the search for vaccine candidates against this economically important disease.

We are greatly thankful to reviewer #1 for the encouraging comments

line 253: should say H31-H34.

Corrected as suggested line: 261

Comments on the Quality of English Language:

The language would benefit from copy-editing by a native speaker. Although it is understandable and mostly reads well, there are consistent grammatical mistakes throughout, including unnecessary use of "the" and use of the incorrect tense, e.g. line 39 should say "evaluate", and multiple times the word "consisted" is used rather than "consisting" etc.

The manuscript text was subjected to English editing for the grammatical mistakes.

Reviewer 2 Report

I read the paper with interest. In view of the development of an effective vaccine against Babesia parasite it is critical to thoroughly understand the (genetics) structure of antigens that can be used. In the current paper, the authors focused on the genetic variability of MSA. They based on data available in GenBank.

I have the following comments and suggestions that may be considered by the authors during revision:

Abstract:

It's too long and too detailed in my opinion. L24-34 can be rewritten and shortened. There is no need to provide such detailed results in this section (e.g. „This was evidenced by the presence of three conserved amino acid repeats in only out of the 199 MSA-1 isolates tested”.).

I also believe that there is no need to include the results of the statistical test in this section. it is enough to mention whether the result was significant or not.

A concisely written abstract will encourage the reader to read the article further.

Introduction:

The introduction is well written and the objective is correctly formulated.

Methods:

L80 Babesia bovis

the description of phylogenetic methods is correct. However, I would suggest explaining the methodology of selecting the model of phylogenetic trees. And why did the authors decide on ML?

Results:

The results are well presened.

Fig 3 and 5 did not contain hoplotyoples numbers

Supplementary:

Fig S1 and S2 did not contain description.

Author Response

Response to the Comments from the editor and reviewers

Manuscript ID: genes-2593345

Title: Genetic diversity of merozoite surface antigens in the global Babesia bovis populations

Comments from the editors and reviewers: 

First, we thank the reviewers for providing constructive comments and suggestions, which were useful for improving the quality of the manuscript. We have now modified our manuscript in acknowledgement of all of the reviewers’ comments and suggestions. In the following pages are our point-by-point responses to each of the comments. The changes made to the manuscript are marked up using the “Track Changes” function. All references are relevant to contents of the revised version of the manuscript.

Reviewer-2

Comments and Suggestions for Authors:

I read the paper with interest. In view of the development of an effective vaccine against Babesia parasite it is critical to thoroughly understand the (genetics) structure of antigens that can be used. In the current paper, the authors focused on the genetic variability of MSA. They based on data available in GenBank.

We are greatly thankful to reviewer #2 for the encouraging comments

I have the following comments and suggestions that may be considered by the authors during revision:

Abstract:

It's too long and too detailed in my opinion. L24-34 can be rewritten and shortened. There is no need to provide such detailed results in this section (e.g. „This was evidenced by the presence of three conserved amino acid repeats in only out of the 199 MSA-1 isolates tested”.).

The abstract was revised and shortened.

The sentence “This was evidenced by the presence of three conserved amino acid repeats in only out of the 199 MSA-1 isolates tested” was deleted.

I also believe that there is no need to include the results of the statistical test in this section. it is enough to mention whether the result was significant or not. A concisely written abstract will encourage the reader to read the article further.

The abstract was revised and rewritten concisely.

Introduction:

The introduction is well written and the objective is correctly formulated.

Thanks

Methods:

L80 Babesia bovis

Corrected as suggested line: 82

The description of phylogenetic methods is correct. However, I would suggest explaining the methodology of selecting the model of phylogenetic trees.

The nucleotide substitution models with the best fit to the data set were selected using the MEGA6 software. Among different models, Hasegawa-Kishino-Yano (HKY) Model showed lower BIC values in all genes fragments.

And why did the authors decide on ML?

We decided to use the ML analysis because it is slightly better than other methods when the evolutionary rate varied drastically among branches.

Results:

The results are well presented.

Thanks

Fig 3 and 5 did not contain haplotypes numbers

We added the haplotypes numbers to figures 3 and 5. 

Supplementary: Fig S1 and S2 did not contain description.

We added the figure captions in the supplementary materials

Reviewer 3 Report

The manuscript is interesting and the authors made a good job as they provide a comprehensive analysis of diversity of B.bovis MSA gene diversity.

The study is well described and the results are presented in detail. Sometimes, the reading is a little too heavy for excessive technical details.

I couldn't find the caption for figures S1 and S2

Author Response

Response to the Comments from the editor and reviewers

Manuscript ID: genes-2593345

Title: Genetic diversity of merozoite surface antigens in the global Babesia bovis populations

Comments from the editors and reviewers: 

First, we thank the reviewers for providing constructive comments and suggestions, which were useful for improving the quality of the manuscript. We have now modified our manuscript in acknowledgement of all of the reviewers’ comments and suggestions. In the following pages are our point-by-point responses to each of the comments. The changes made to the manuscript are marked up using the “Track Changes” function. All references are relevant to contents of the revised version of the manuscript.

Reviewer-3

Comments and Suggestions for Authors

The manuscript is interesting and the authors made a good job as they provide a comprehensive analysis of diversity of B. bovis MSA gene diversity.

We are greatly thankful to reviewer #3 for the encouraging comments

The study is well described and the results are presented in detail. Sometimes, the reading is a little too heavy for excessive technical details.

The manuscript was revised accordingly.

I couldn't find the caption for figures S1 and S2

We added the figure captions in the supplementary materials
